# The Diversity of Fungi Involved in Damage to Japanese Quince

**DOI:** 10.3390/plants11192572

**Published:** 2022-09-29

**Authors:** Inta Jakobija, Biruta Bankina, Alise Klūga, Ance Roga, Edmunds Skinderskis, Dāvids Fridmanis

**Affiliations:** 1Faculty of Agriculture, Latvia University of Life Sciences and Technologies, Liela Street 2, LV-3001 Jelgava, Latvia; 2Institute of Plant Protection Research “Agrihorts”, Latvia University of Life Sciences and Technologies, Paula Lejina Street 2, LV-3001 Jelgava, Latvia; 3Latvian Biomedical Research and Study Centre, Rātsupites Street 1 k-1, LV-1067 Riga, Latvia

**Keywords:** *Chaenomeles japonica*, symptoms, *Botrytis*, *Monilinia*, ecological niches

## Abstract

In recent years, Japanese quince (*Chaenomeles japonica*) plantations in Latvia have increased. Interest in breeding Japanese quince is also known in other European countries and Russia. The occurrence and harmfulness of fungal diseases have become increasingly significant. However, there is a lack of overall information in the literature about the diversity of fungi afflicting *C. japonica*. In our study, we aimed to determine the diversity of fungi associated with *C. japonica* in Latvia, with the possibility of identifying the fungi that are most characteristically associated with certain parts of the plant. Our research was conducted from 2017 to 2019 in eight Japanese quince plantations in Latvia. Samples of plant parts with disease symptoms were collected. Pure cultures of fungi were obtained and identified using ITS region sequencing. We determined the relative density of identified genera of fungi, which were grouped using hierarchical cluster analysis depending on the plant part from which they were found. Various disease-like symptoms were observed and described. A total of 538 isolates of fungi were obtained that belong to 36 genera and represent different ecological niches. *Fusarium*, *Alternaria*, *Botrytis*, and *Sarocladium* were the genera most frequently found during our study. The number of identified cases of fungal genera differed depending on the part of the plant from which the fungi were obtained. However, it is not possible to relate a specific genus of fungus to only one certain part of a plant. Further research is needed to clarify the pathogenicity of detected fungi and the composition of species in the detected genera of fungi.

## 1. Introduction

Japanese quince (*Chaenomeles japonica*) belongs to the Rosaceae family and is considered a nutrient-rich raw material in the food industry due to its bioactive ingredients, its aroma, and the high fibre content of its fruits [1,2]. Furthermore, the byproducts of Japanese quince processing are recognized for their promising use in cosmetology and medicine [3]. In Latvia, the breeding of Japanese quince for fruit production began in the 1950s [4]. In 1993, plantations of Japanese quince in Latvia and Lithuania covered more than 400 ha [5]. In recent years, the area of quince plantations in Latvia has increased, reaching approximately 500 ha. Several studies on the possibilities for quince use and breeding have been conducted in Estonia [6] and many other countries, such as Sweden [7], Poland [8], Bulgaria [9], Ukraine [10], Belarus [11], and Russia [12,13].

Japanese quince has long been considered a plant unaffected by serious diseases. However, as the planting areas of Japanese quince continue to grow, there is the potential for increased disease occurrence and harmfulness. Previous studies in Latvia have shown that fruit rot has reached agronomically significant levels in some growing seasons [14], with a high incidence of leaf spot diseases observed in several plantations [15]. Moreover, the presence of diseases also causes problems in other quince-growing countries, as indicated by research results where various fungi have been detected in Japanese quince [16,17,18]. Some of these fungi are well-known plant pathogens: *B. cinerea* has been detected and isolated from *C. japonica* in Sweden [5,16], Lithuania [17], and Russia [18]; *Monilinia fructigena* was found on *C. japonica* fruits in the Moscow region [18] and Sweden [16]; *M*. *cydonia* has been found to damage flowers in Russia [18], and *M. linhartiana* was found to be the pathogen responsible for fruit rot of *Cydonia oblonga* in Spain [19].

In many cases, the ecological niche of fungi associated with Japanese quince is unclear. *Fusarium* and *Alternaria* spp. are recognized as pathogens and components of the endophytic mycobiota of Japanese quince [16,18], *Cladosporium* spp. have been isolated from *C. japonica* in Sweden and Lithuania [16,17] and are considered endophytes in Russia [18] and *Penicillium* spp. have been detected in *C. japonica* in Sweden, Russia, and Lithuania [16,17,18].

There is a lack of overall information in the literature about the diversity of fungi in *C. japonica*, particularly regarding pathogens. We aimed to determine the diversity of fungi in *C. japonica* in Latvia and possibly discover which fungi are most characteristic of certain parts of the plant.

Knowledge about the possible damage caused by diseases and fungal composition in *C. japonica* may help in selecting the most appropriate research methods for further studies on fungal identification and plant protection against diseases.

## 2. Results

Various spots were found on the leaves of *C. japonica* (Figure 1). They were small and black with a purple or red halo (diameter approx. 1–2 mm) (Figure 1c), and dark brown with or without round or irregularly shaped concentric rings (diameter approx. 0.5–1 cm) (Figure 1a,b) and red fuzzy spots (Figure 1d).

Brown tops of leaves with concentric bands (Figure 1a) and brown or chlorotic edges of leaves were also observed.

Spots were more clearly visible on the adaxial surface of the leaf but also noticeable from the abaxial side.

We also observed overwintered mummified fruits with various external features such as dark or light brown and black, difficult-to-deform or easily deformable, sometimes with a paper-like skin.

Small, irregular red, purple, brown, or black spots with a pale centre, in most cases rounded with a red halo, were the most typical examples of damage observed on fruit (Figure 2a). Fruit rot was also observed, but the symptoms varied. The rotted areas on fruits were soft and light to dark brown, sometimes with concentric darker and lighter coloured bands. In wet conditions, the surface of rotten fruit was sometimes found to be covered with grey mycelium or cream-coloured mycelium aggregations (Figure 2b–e).

Areas of detected shoot damage were round or oval with dead, sunken, or cracked bark (Figure 3b–d). We also observed that the bark of the damaged shoots peeled off from the damaged areas (Figure 3a).

During our studies, a total of 538 fungal isolates were obtained from *Chaenomeles japonica* and identified at the genus level. These obtained isolates correspond to 36 genera of fungi. The relative density exceeded 2% for only 10 genera, with the majority rarely found based on incidence (Figure 4).

*Fusarium* spp., *Alternaria* spp., *Botrytis* spp., and *Sarocladium* spp. were the dominant genera, with a relative density of 21%, 15%, 13%, and 9%, respectively. Fungi of the genera *Clonostachys, Cladosporium, Epicoccum, Trichoderma, Monilinia,* and *Boeremia* were identified often (relative density ranging from 3% to 6%). The relative density of other genera did not exceed 2%.

The diversity of fungi varied according to the plant part. A total of 256 isolates were obtained from leaves, 202 from fruits, 74 from shoots, and only 6 from inflorescences, representing 19, 28, 17, and 4 fungal genera, respectively.

In order to determine which genera of fungi are associated with particular plant parts, the identified genera were divided into five different clusters depending on the frequency of their isolation from a particular damaged part of *C. japonica* (Figure 5).

*Fusarium* spp. and *Botrytis* spp. (Cluster 1) were isolated from inflorescences, shoots, fruits, and leaves (cluster centroids 0.5, 41.0, 33.0, and 17.0, respectively). These fungi were detected in all parts of *C. japonica* but rarely found in inflorescences. *Fusarium* spp. were equally isolated from leaves and fruits, though more frequently compared to the proportional distribution of the isolates obtained from shoots, whereas representatives of the genus *Botrytis* were isolated from fruits more frequently than from leaves and shoots.

Fungi from the genera *Alternaria* and *Sarocladium* (Cluster 2) were rarely isolated from inflorescences (only one case of *Alternaria*), shoots, and fruits and most often from leaves (depending on cluster centroids of inflorescences, fruits, leaves, and shoots—0.5, 7.0, 52.5, and 4.5, respectively). These fungi were found to be most associated with the infected leaves of *C. japonica*.

*Clonostachys*, *Trichoderma*, and *Monilinia* (Cluster 3) were isolated mainly from fruits and, in some cases, from inflorescences, leaves, and shoots (depending on cluster centroids of inflorescences, fruits, leaves, and shoots—0.3, 16.3, 4.7, and 3.0 respectively). These fungi can be described as relatively common in *C. japonica* fruits. In particular, *Clonostachys* spp. were isolated from leaves more often than from shoots and inflorescences. *Monilinia* spp. were isolated from shoots more often than from leaves. However, the occurrence of *Trichoderma* spp. on leaves and shoots was equal.

Representatives of Cluster 4 were rarely isolated from fruits and shoots but relatively often from *C. japonica* leaves and not from inflorescences (depending on cluster centroids of inflorescences, fruits, leaves, and shoots—0.0, 3.4, 11.4, and 1.0, respectively) (Figure 5). This group included *Cladosporium*, *Epicoccum*, *Boeremia*, *Arthrinium*, and *Akanthomyces* spp.— fungal genera that were mainly associated with leaves. Moreover, *Epicoccum* and *Akanthomyces* spp. were not found in shoots. *Cladosporium* and *Boeremia* species were isolated from shoots less frequently than from leaves and fruits, whereas *Arthrinium* spp. were found in shoots more often than in fruits but significantly less than in leaves.

In general, the majority (24 out of the total 36) of identified genera of fungi (Cluster 5) were characterized by a small number of identified cases (depending on cluster centroids of inflorescences, fruits, leaves, and shoots—0.1, 1.7, 0.6, and 0.7, respectively), and each genus was found only in one or two parts of *C. japonica* plants. For this reason, it was impossible to determine which plant parts were most susceptible to certain fungi.

## 3. Discussion

Different fungi were isolated from similar disease-like symptoms; therefore, it was impossible to link them with specific fungi. Similar results were obtained by Norin and Rumpunen (2003), who noted that damaged areas on the leaves of *C. japonica* varied in shape, size, and colour and found no correlation between disease signs and the fungi isolated from leaves [16].

Small brown or black spots with a red or purple halo on leaves, approximately 1–2 mm in diameter, were observed in our study. Similar signs on leaves were also described in research conducted in Russia [18]. Interestingly, we detected previously undescribed disease symptoms in quince leaves: red fuzzy spots (approximately 1 cm in diameter) and brown edges.

Small red, brown, and black spots with or without a pale centre have been noticed on Japanese quince fruits in other countries [16,17,18], which is similar to the findings in our study.

Reports of brown concentric fruit rot are common in *C. japonica* research in Sweden [16], which appear similar to our results where rotted areas with darker and lighter concentric bands on quince fruits were observed. Detection of a brown-to-yellow set of conidia on the surface of *C. japonica* fruits has been reported in Russia [18], with the same phenomenon observed in the present study.

We observed the wilting of buds and leaves and/or the damaged woody parts developing into bark necroses, similar to the symptoms detected on Japanese quince shoots in other countries [16,18].

Various disease-like symptoms were observed in different parts of Japanese quince in the present research. However, there is a general lack of information in the literature describing the disease symptoms of Japanese quince; only a few information sources could be found, suggesting that it is necessary to describe the symptoms of Japanese quince diseases in detail.

The genera *Fusarium* and *Alternaria* were the most frequently found of all the obtained isolates and were mentioned in other studies on Japanese quince [16,17,18]. Fungi of these genera represent a wide range of saprotrophs and well-known plant pathogens in different hosts, including plants from the Rosaceae family [20,21,22,23,24]. The exact species of *Fusarium* and *Alternaria* have not been determined yet, and precise identification and detailed research are required to clarify the ecological niches of the isolated fungi. There are a very few data regarding the spectrum of *Fusarium* and *Alternaria* fungi in the literature; therefore, it is important to continue studies on fungi associated with *C. japonica*.

*Botrytis cinerea* is the fungus most frequently found in Japanese quince plantations in Sweden [16], which is similar to our research results. The relative density of the genus *Monilinia* (4%) was comparably small but considerable in the present study. *Monilinia fructigena* was the fungus most commonly found in *C. japonica* collection plantations in Moscow during the 2010–2017 monitoring [18], contrasting our findings. Several *Monilinia* species have also been reported in *C. japonica* in Sweden [16]. Furthermore, the genus *Neofabraea*, although rarely detected in our study, has been found in Japanese quince in other countries [16,17,18]. Considering that the aforementioned genera included several well-known fungi species pathogenic to Rosaceae plants [22,23,24,25,26,27,28,29,30], it will be important to clarify the composition of these species in Japanese quince.

In our study, the relative density of *Epicoccum*, *Trichoderma*, and *Cladosporium* was low (4% for each genus); *Arthrinium*, *Aureobasidium*, *Penicillium*, *Trichothecium*, and *Didymella* were also rarely isolated. In other countries, these genera have also been obtained from *C. japonica* [16,17,18]. There have also been reports of this species’ association with other Rosaceae plants [20,23,31,32,33,34]. *Pseudopithomyces*, *Truncatella*, *Trametes*, *Rhizoctonia*, *Nigrospora*, and *Rosellinia* were rarely detected in our study and were not mentioned previously in direct connection with *C. japonica*. However, some reports indicate that they are common to other Rosaceae plants—stone fruits, pome fruits, and roses [20,35,36,37,38]. The role of the aforementioned genera in the phytosanitary condition of Japanese quince is unclear due to their endophytic and saprotrophic properties [18,39,40,41,42,43,44], which would be interesting to discover.

Although there is no information in the literature indicating that *Clonostachys* spp. have been detected in *C. japonica* in other countries, this fungal genus was found in 6% of cases in our study. The possibility of interconnecting *Clonostachys* spp. with *C. japonica* is supported by research conducted on plum diseases in Latvia [45]. In addition, it should be mentioned that *Clonostachys rosea* (formerly *Gliocladium roseum*) and other *Clonostachys* spp. known as biocontrol agents of various fungal pathogens [44,46,47] including *Botrytis cinerea* [48].

*Sarocladium* was among the most common fungal genera in our study; however, no information could be found in the literature about the occurrence of representatives of this genus in *C. japonica*. *Sarocladium liquaenensis* and *S. mali* have been reported as causal agents of brown spots on apples in China [49]. On the other hand, there are species with endophytic [50,51] and saprotrophic traits [52] among the species of *Sarocladium*. It would be valuable to determine the role of *Sarocladium* in *C. japonica* under phytosanitary conditions.

Fungi from the genera *Boeremia*, *Diaporthe*, *Discosia*, *Coniophora*, *Talaromyces*, *Allantophomopsis*, *Neoascochyta*, *Sordaria*, *Simplicillium*, *Stagonosporopsis*, and *Pestalotiopsis* were rarely isolated and their detection in *C. japonica* has not been previously reported. It should also be noted that these species have not previously been reported as associated with Rosaceae plants.

According to the cluster analysis results, the genera *Fusarium* and *Botrytis* were detected in all parts of *C. japonica*, in agreement with the data of other research reports [16,17,53,54].

Fungi from the genera *Alternaria* and *Sarocladium* were found mainly associated with infected *C. japonica* leaves, but in rare cases, they were also obtained from other parts of the plant. *Alternaria* spp. are found associated with *C. japonica* leaves in Russia [18], in accordance with our findings. Nevertheless, *Sarocladium* spp. have been associated with plant parts similar to those observed in our study, but only in other plants, for example, on apple fruits in China [49] and strawberry leaves in Argentina [55].

Fungi from the genera *Clonostachys*, *Trichoderma*, and *Monilinia* were mainly found in *C. japonica* fruits, and these results coincide with research findings in other countries [16,56,57].

A group of fungal genera, namely *Cladosporium*, *Epicoccum*, *Boeremia*, *Arthrinium*, and *Akanthomyces*, were characteristic mainly of quince leaves, but in rare cases, they were also isolated from fruits and shoots. The occurrence of *Cladosporium* spp. in Japanese quince flowers appears common based on investigations in Sweden [16], contrary to results obtained in Latvia, where these fungal genera were mainly detected in leaves but not flowers. This difference may be explained by different compositions of species within the isolated representatives of the genus *Cladosporium* in Sweden and Latvia. In Sweden, *Epicoccum* spp. are associated with *C. japonica* leaves [16].

The remaining fungal genera comprised Cluster 5, where each genus was found only in one or two parts of *C. japonica*. For instance, *Penicillium* species were isolated only from fruits of *C. japonica* in this study, whereas in Sweden, they were isolated from petals and leaf spots and also detected on *C. japonica* fruits [16]. The presence of *Neofabraea* in fruits was self-evident due to its well-known properties as a causal agent of fruit rot in apples and pears [58]. As in our study, *Trichothecium* species have been detected in *C. japonica* fruits in Lithuania [17]. On leaf samples obtained in Latvia by Norin and Rumpunen (2003), *Septoria cydoniae*, *Phoma pomorum* (current name: *Didymella pomorum*), *Asteromella* spp., and *Ramularia* spp. were detected, from which only *Didymella* was isolated from *C. japonica* leaves and shoots in our study [16].

The diversity of pathogenic fungi associated with *C. japonica* disease symptoms was determined. Our results indicate that the fungi isolated from *C. japonica* are mostly the same as those detected in other plants belonging to the Rosaceae family. Several representatives of identified genera detected in *C. japonica* include pathogens from other plant families and different saprotrophs and endophytes. Some of the identified genera were even recognized insect parasites. The majority of the 36 identified fungal genera belonged to the phylum Ascomycota, with the exception of two genera belonging to Basidiomycota.

In order to ascertain whether the fungi obtained from *C. japonica* plants cause disease symptoms and are not secondary pathogens, endophytes, or saprotrophs, further research is required. To better understand their biological peculiarities, in-depth molecular genetic analyses are also required to identify the obtained fungal genera at the species level. The possibility of linking disease symptoms with specific genera of fungi proved to be very limited and, in most cases, impossible because different fungal genera were obtained from similar disease-like symptoms.

## 4. Materials and Methods

Samples of *C. japonica* fruits, leaves, shoots, and inflorescences with disease symptoms were collected from eight plantations in Latvia in the 2017–2019 vegetation seasons. Observations were conducted every two weeks during all vegetation periods.

A 1.25% solution of sodium hypochlorite was used for sample tissue sterilization. The duration of sterilization varied from 30 s for leaves and fruits to 2 min for woody parts. After sterilization, samples were rinsed in sterile water 3 times and plased on a sterile filter paper for drying.

Pieces of the fungi-damaged area of fruits and shoots were placed on on potato dextrose agar (PDA; VWR™, IBI Scientific, Dubuque, IA, USA), but leaf segments with symptoms—on V8 juice agar (M638 Himedia Laboratories, Einhausen, Germany). Both media were enriched with 100 ppm L^−1^ streptomycin sulphate salt (Sigma–Aldrich, Burlington, MA, USA). Plates incubated at the temperature 22–23 °C in dark till colonies of fungi appeared. Then, from each colony plugs (3–5 mm) were placed on culture media with colony side down and afterwards were purified by fungal tip isolation. Pure cultures were incubated at 22 °C in darkness for one or two weeks depending on growing rate.

The obtained fungal isolates were grouped by morphological characteristics and examined with a microscope when necessary; one isolate from each group was selected for ITS region sequencing.

DNA was extracted from fresh, pure fungal cultures using E.Z.N.A.^®^ HP Fungal DNA Kit (Omega BIO-TEK, Norcross, GA, USA). The fungal tissue material for extraction was cut out of a plate (~1 cm^2^, ≤200 mg) and transferred into a 2 mL microcentrifuge tube containing ~20 0.5 mm glass beads (Qiagen, Germantown, MD, USA). Samples were ground using TissueLyser II (Qiagen, Germantown, MD, USA) for up to 8 min. After grinding, 500 µL of CSPL buffer was added to the samples followed by vortexing for 8 min using TissueLyser II. Then, 10 µL of 2-mercaptoethanol (VWR™, IBI Scientific, Dubuque, IA, USA) was added to the samples and again vortexed using TissueLyser (8 min). To remove RNA, 5 µL of 10 mg/mL RNaseA (Omega BIO-TEK, Norcross, GA, USA) was added. Samples were briefly vortexed (Elmi-Tech, Riga, Latvia) and incubated in a dry block heating thermostat (Thermo Block TDB-120, BIOSAN, Riga, Latvia) for 30 min at 37 °C. Furthermore, the sample was incubated at 65 °C for 15 min. To mix the sample, sample tubes were inverted at least twice during the incubation. After the incubation, 800 µL of chloroform/isoamylalcohol (24:1) (ACROS Organics™, Geel, Belgium) was added, and the samples were briefly vortexed to mix thoroughly. Samples were centrifuged at 10,000× *g* (SIGMA^®^ 1–14, Roedermark, Germany) for 5 min.

After centrifugation, 300 µL of supernatant was collected in a new 1.5 mL centrifugation tube, and 150 µL of CXD buffer and 300 µL of cold 96% ethanol were added. The sample was vortexed, obtaining a homogeneous mixture which was transformed into a HiBind^®^ DNA mini-column. Samples were centrifuged at 10,000× *g* for 1 min. DNA was washed twice with 650 µL of DNA wash buffer, followed by centrifugation at 10,000× *g* for 1 min each time. To remove excess DNA wash buffer, empty columns were centrifuged at 14,000× *g* for 2 min. HiBind^®^ DNA mini-columns were transferred into new 1.5 mL centrifuge tubes, and 25 µL of sterile deionized water heated to 65 °C was added to the centre of the columns followed by centrifugation at 14,000× *g* for 1 min to elute DNA from the column. DNA elution was repeated, resulting in 50 µL of eluted DNA sample. The DNA concentration was detected using NanoDrop^®^ ND-1000 spectrophotometer (ThermoFisher Scientific, Waltham, MA, USA). The samples were stored at −20 °C until further DNA amplification.

The PCR amplification reaction included 10 µL MyTaq™HS Red Mix (Bioline, London, UK), 0.4 µL (20 mM) each of forward (ITS1-F) and reverse (ITS4) primer, 8.2 µL nuclease-free water (ThermoFisher Scientific, Waltham, MA, USA), and 1 µL fungal DNA in a final reaction volume of 20 µL. The following amplification conditions were used: initial denaturation at 95 °C for 1 min, followed by 35 cycles of denaturation at 95 °C for 15 s, annealing at 51 °C for 15 s, and extension at 72 °C for 10 s. The final extension was performed at 72 °C for 8 min, with a hold temperature of 10 °C. PCR amplifications were conducted using a GeneAmp^®^ PCR System 9700 (Applied Biosystems, Waltham, MA, USA).

The quality of PCR products was tested using 1% TopVision Agarose gel (ThermoFisher Scientific, Waltham, MA, USA) before sequencing. Unpurified PCR products were sent for Sanger sequencing to Latvia Biomedical Research and Study Center (Riga, Latvia).

The received PCR products were enzymatically treated to remove the excess of dNTPs and primers (0.5 µL Exonuclease I (Thermo Fisher Scientific, Waltham, MA, USA), 2 µL Shrimp Alkaline Phosphatase (Thermo Fisher Scientific, Waltham, MA, USA), incubated at 37 °C for 40 min and at 95 °C for 20 min) and 1 µL of thus acquired fragment solution was transferred to BigDye^®^ Terminator v3.1 Cycle Sequencing reaction mixture (both from Applied Biosystems, Waltham, MA, USA), which was prepared according to manufacturer’s instructions, supplemented with 1 µL of 5mM ITS1-F (forward) or ITS4 (reverse) primer and afterwards incubated in thermal conditions that were recommended by manufacturer using a GeneAmp^®^ PCR System 9700. Acquired sequencing products were then analyzed on 3130xl Genetic Analyzer (Applied Biosystems, Waltham, MA, USA) and resulting data was manually inspected using FinchTV chromatogram viewer (v. 1.5.0, Geospiza, Inc., Seattle, WA, USA). Afterwards sequence reads of both strands were aligned using MEGA11 software (v. 11.0.13, MEGA Software Development Team, Tokyo, Japan & Philadelphia, PA, USA) [59] and resulting coting was BLAST^®^ searched against NCBI nucleotide database to identify taxonomic source of record with highest sequence identity (https://blast.ncbi.nlm.nih.gov/Blast.cgi, assessed 1 May 2022).

The relative density of the identified fungal genera was determined as the proportion of a particular genus among all the obtained isolates. Identified fungal genera were classified according to the plant parts from which they were isolated using cluster centroids (multidimensional average of the cluster) [60] calculated by Hierarchical Cluster analysis (method Ward, distance Euclidean). Program R version 4.1.1 was used for the analysis.

## 5. Conclusions

Different fungi were isolated from similar disease symptoms, suggesting that a particular type of damage was most likely caused by a complex of fungi.

A high diversity of fungi, a total of 36 fungal genera, was detected in the damaged parts of *C. japonica* plant. The highest diversity of fungi, 28 genera, was found in fruits.

The most frequently identified species were *Fusarium*, *Alternaria*, *Botrytis*, and *Sarocladium*. The relative densities of the genera *Clonostachys*, *Cladosporium*, *Epicoccum*, *Trichoderma*, *Monilinia*, and *Boeremia* were slightly lower but considerable.

The number of identified cases of fungal genera differed depending on the part of the plant from which the fungi were obtained. However, it is impossible to relate a specific genus of fungi to certain part of a plant only.

## Figures and Tables

**Figure 1 plants-11-02572-f001:**
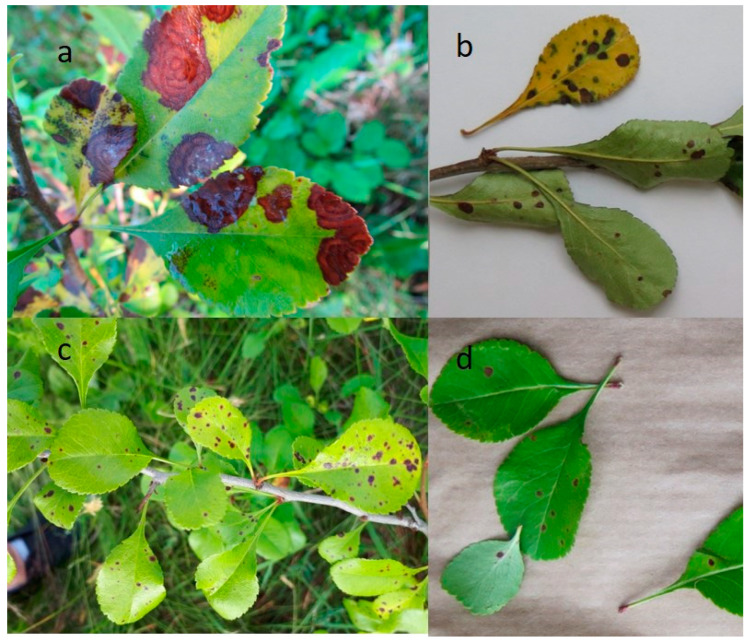
Various leaf spots on Japanese quince leaves: (**a**) brown irregular spots with concentric rings; (**b**) dark brown necrotic spots; (**c**) small dark spots with red halo; (**d**) small fuzzy spots.

**Figure 2 plants-11-02572-f002:**
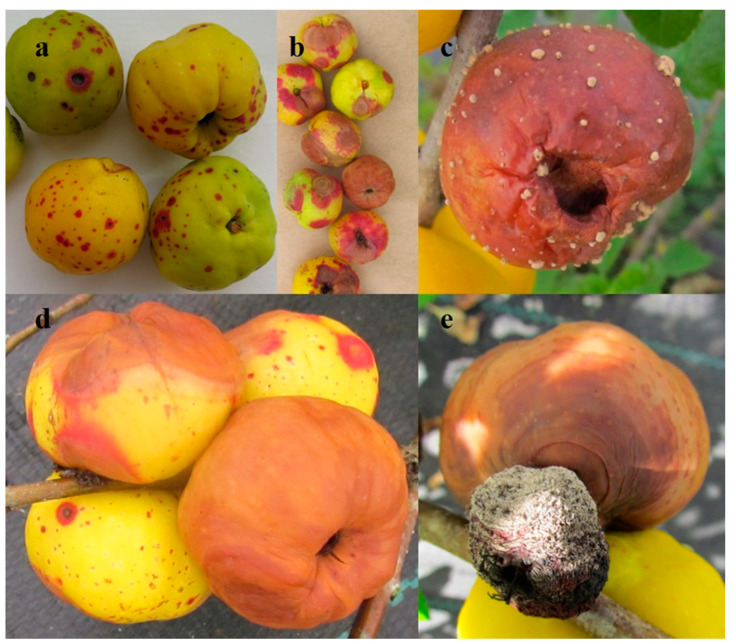
Damage to Japanese quince fruits: (**a**)—fruit spots; (**b**–**e**): various symptoms of fruit rot.

**Figure 3 plants-11-02572-f003:**
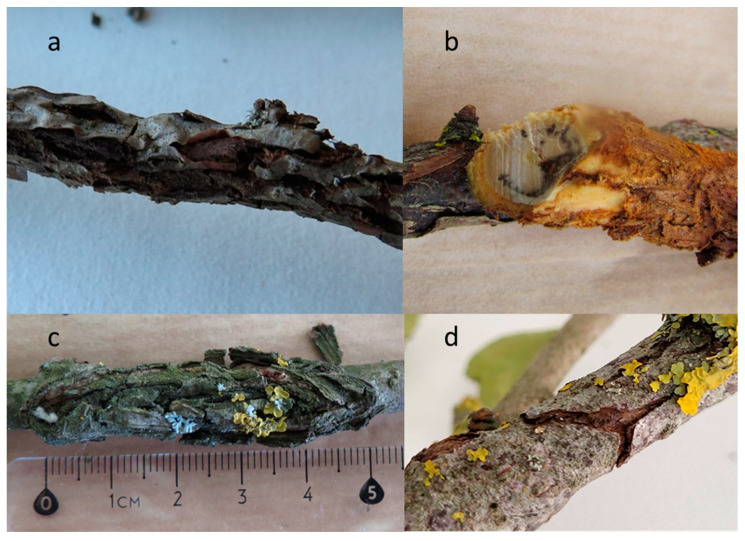
Damage to Japanese quince shoots: (**a**) the bark peeled off from the damaged area; (**b**) damaged shoot in cross section; (**c**) oval wound with dead bark; (**d**) cracked bark.

**Figure 4 plants-11-02572-f004:**
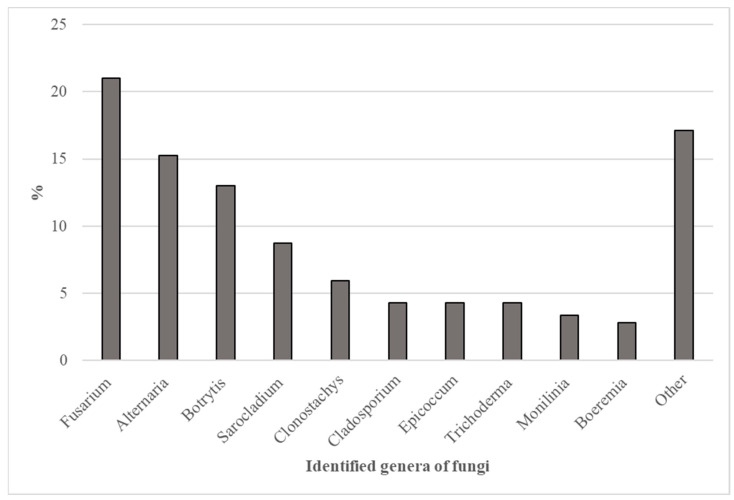
Relative density of the fungal genera obtained from *C. japonica*. Other includes *Arthrinium, Diaporthe, Coniophora, Talaromyces, Akanthomyces, Didymella, Penicillium, Allantophomopsis, Aureobasidium, Neoascochyta, Sordaria, Neofabraea, Discosia, Pseudopithomyces, Truncatella, Trichothecium, Rhizoctonia, Trametes, Stagonosporopsis, Pestalotiopsis, Nigrospora, Rosellinia, Hormographiella,* and *Isaria*.

**Figure 5 plants-11-02572-f005:**
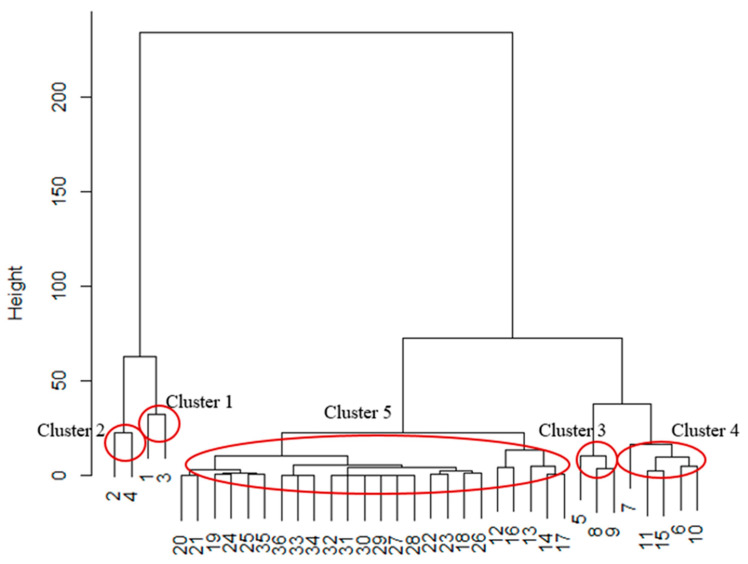
Dendrogram of identified clusters of fungal genera based on several isolation cases from different parts of *C. japonica.* The numbers in dendrogram denote the genera of fungi: 1—*Fusarium*, 2—*Alternaria*, 3—*Botrytis*, 4—*Sarocladium*, 5—*Clonostachys*, 6—*Cladosporium*, 7—*Epicoccum*, 8—*Trichoderma*, 9—*Monilinia*, 10—*Boeremia*, 11—*Arthrinium*, 12—*Diaporthe*, 13—*Coniophora*, 14—*Talaromyces*, 15—*Akanthomyces*, 16—*Didymella*, 17—*Penicillium*, 18—*Allantophomopsis*, 19—*Aureobasidium*, 20—*Neoascochyta*, 21—*Sordaria*, 22—*Simplicillium*, 23—*Neofabraea*, 24—*Discosia*, 25—*Pseudopithomyces*, 26—*Truncatella*, 27—*Trichothecium*, 28—*Rhizoctonia*, 29—*Trametes*, 30—*Crustomyces*, 31—*Stagonosporopsis*, 32—*Pestalotiopsis*, 33—*Nigrospora*, 34—*Rossellina*, 35—*Hormographiella*, and 36—*Isaria*.

## Data Availability

All data are included in the main text.

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
