# Peer review of "The Diversity of Fungi Involved in Damage to Japanese Quince"

_plants, 2022, doi:10.3390/plants11192572_

Round 1

Reviewer 1 Report

This is an original research study, well planned, well performed and well reported. Literature is well covered and properly cited. There are only a few editorial changes to be made in the manuscript (latin genera and species mentioned to be in italics line 117, 152, 216, 219, 220, 247, 269, 354) (a space missing line 285: 10 mg).

Author Response

This is an original research study, well planned, well performed and well reported. Literature is well covered and properly cited.

There are only a few editorial changes to be made in the manuscript (latin genera and species mentioned to be in italics line 117, 152, 216, 219, 220, 247, 269, 354) (a space missing line 285: 10 mg).

Thank reviewer for careful reading of our work. We have made necessity corrections.

Response 1: Lines 117, 152, 216, 219, 220, 247, 269, 354 – corrected.

Response 2: A space missing line 285: 10 mg – corrected

Reviewer 2 Report

My overall comment is this manuscript needs to be readdressed and work repeated to be more comprehensive.  Other comments are in the attached file.  

Author Response

Thanks to the reviewer for careful reading of our work and many excellent ideas.

The reviewer has given us several good ideas for further studies, we have continued studies and we have started to prepare the next manuscripts. There is a lack of information about the fungi associated with quinces. We believe that basic knowledge about the fungal spectrum is necessary to conduct more detailed research.

We have made the required corrections and answered the questions.

For example, were there differences in the fungi you identified between the different plantations?  Were there differences between the years? 

Response 1. Our aim of this research was to determine the spectrum of fungi associated with damaged quinces, because there are only a few publications about quinces, their pathogens and other fungi.

You indicate the samples were obtained from "plant parts with disease symptoms" (line 17).  The main issue with your sampling and processing procedure it you still do not know whether the fungi you have culture as the cause of the disease symptoms. 

Response 2.  It was the first attempt to identify the fungi associated with the damages of C. japonica. Therefore, this study was intended to provide basic knowledge for future studies of this type in C. japonica, including the causal agents of each type of the symptoms.

The fundamental issue with the manuscript is your preparation of samples to obtain a pure culture is flawed.  What your study has provided is epiphytic fungi only and more than likely represents saprophytic fungi, rather than pathogenic fungi. 

The authors should have surface sterilised material prior to plating, which is the standard practice in plant pathology. 

Response 3.

Thanks for the remark! We have made a mistake not mentioning the method of sterilization; now we have corrected it and described the method in the text.

Of course, we have obtained fungi from different ecological quilts – including saprotrophs, pathogens and possible endophytes.

The authors give no indication whether single spore culturing or hyphal tipping was performed to obtain the one culture used for sequencing. 

Response 4.

Hyphal tipping was conducted to obtain pure cultures – the necessary text is added.

Lines 269 to 277 need to be expanded and explained in greater detail.  This leads to the next major issue whereby the ITS sequencing was with a forward primer only (ITS1-F - Line 317).  The reverse sequence should have been undertaken to produce a contig and then the consensus blasted for fungal genera identity. 

Response 5.

The text was expanded, and methods were explained in detail.

The authors have indicated there was a high diversity of fungi with 36 genera indicated.  This was only from 538 isolates (over a three year period from 8 plantations).  This a relative low number of isolates for such a study.  Additionally, was the sampling undertaken as different time points in the growing season?  It is important to put these details as it will affect the subsequent isolation of fungi. 

Response 6. We have only 538 fungal isolates, because we collected only symptomatic tissues and we were interested only in fungal damages. The time of samples collecting is added in the text.

You results section need more information.  Indicate what the observation was on the adaxial and abaxial surface of the leaf (line 66-69).  What was the average size of the lesions observed, comparative to the leaf.  This detail is missing. 

Response 7. The required text was added.

Lind 76 you indicate the damage observed on fruit, but provide not additional information regarding whether they fruit from the start or the season of need the end.  Information regarding the Brix would have been useful.

Response 8. The purpose of this study was to understand the spectrum of fungi, which will be the base for further studies, including changes in the fungal spectrum depending on ripening stages.

The authors indicate in wet condition’s (Line 79) what some of the symptoms were like, but how many samples were obtained in such conditions?  Were there genera differences with these fruit to others?  Further details need to be provided. 

Response 9. The impact of weather conditions will be evaluated in further studies.

The analysis of dominant genera can be further enhanced by undertaking an analysis looking at plantation, to determine whether this may affect the recovery of the fungi.  Ultimately, to make a statement regarding fungal diversity, the authors needed to look at the diversity on the non-disease areas of the fruit.  That way they could then indicate whether the “diseased” parts of the plant are exclusively associated with the genera isolated or whether there is a general background fungal community.  This has not been attempted in the current study. 

Response 10. The reviewer suggests several excellent ideas for further studies, but they were not the aim of the present research.

Additionally, the authors cannot determine which fungi are pathogenic as 1) isolation was not done from surface sterilised material to isolate from internal tissue and 2) Koch’s postulates were not undertaken to indicate whether the genera isolated are those causing the disease symptoms.

Response 11. These suggestions will be taken into account in our further research.

Reviewer 3 Report

The paper can be accepted in the present form

Author Response

Thanks reviewer for careful reading of our work.

Reviewer 4 Report

The study provides comprehensive research on the diversity of fungal species associated with diseased Japanese quince, and the relationship between plant parts and fungal species is investigated.

1. Authors have described the symptoms in detail, but more specific symptom types are suggested to divided as the second to forth paragraphs in Discussion. And please supplement the legends corresponding to the symptom description to make symptom types clearer.

2. According to the results, the number of isolates from shoots accounted for a high proportion, please provide symptom photos of shoot damage as Figure 1 and Figure 2.

3. Line 102 to 103, “74 isolates were obtained from fruits, representing 74 fungal genera”, one genus only has one isolates? Please confirm that.

Author Response

Thanks reviewer for careful reading of our work. We have made necessity corrections.

Authors have described the symptoms in detail, but more specific symptom types are suggested to divided as the second to forth paragraphs in Discussion.

Response 1: Corrected

And please supplement the legends corresponding to the symptom description to make symptom types clearer.

Response 2: Legends are added in Figure 1 and noted in the text.

According to the results, the number of isolates from shoots accounted for a high proportion, please provide symptom photos of shoot damage as Figure 1 and Figure 2.

Response 3: Figure 3 with legends is added.

Line 102 to 103, “74 isolates were obtained from fruits, representing 74 fungal genera”, one genus only has one isolates? Please confirm that.

Response 4. This was a mistake in the recorded numbers and also swapped “fruits” with “shoots”. Text is corrected.